# Radiomics-Based Detection of COVID-19 from Chest X-ray Using Interpretable Soft Label-Driven TSK Fuzzy Classifier

**DOI:** 10.3390/diagnostics12112613

**Published:** 2022-10-27

**Authors:** Yuanpeng Zhang, Dongrong Yang, Saikit Lam, Bing Li, Xinzhi Teng, Jiang Zhang, Ta Zhou, Zongrui Ma, Tin-Cheung (Michael) Ying, Jing Cai

**Affiliations:** 1Department of Health Technology and Informatics, The Hong Kong Polytechnic University, Hong Kong, China; 2Department of Medical informatics, Nantong University, Nantong 226007, China

**Keywords:** class compactness graph, COVID-19, radiomics, soft label, TSK fuzzy systems

## Abstract

The COVID-19 pandemic has posed a significant global public health threat with an escalating number of new cases and death toll daily. The early detection of COVID-related CXR abnormality potentially allows the early isolation of suspected cases. Chest X-Ray (CXR) is a fast and highly accessible imaging modality. Recently, a number of CXR-based AI models have been developed for the automated detection of COVID-19. However, most existing models are difficult to interpret due to the use of incomprehensible deep features in their models. Confronted with this, we developed an interpretable TSK fuzzy system in this study for COVID-19 detection using radiomics features extracted from CXR images. There are two main contributions. (1) When TSK fuzzy systems are applied to classification tasks, the commonly used binary label matrix of training samples is transformed into a soft one in order to learn a more discriminant transformation matrix and hence improve classification accuracy. (2) Based on the assumption that the samples in the same class should be kept as close as possible when they are transformed into the label space, the compactness class graph is introduced to avoid overfitting caused by label matrix relaxation. Our proposed model for a multi-categorical classification task (COVID-19 vs. No-Findings vs. Pneumonia) was evaluated using 600 CXR images from publicly available datasets and compared against five state-of-the-art AI models in aspects of classification accuracy. Experimental findings showed that our model achieved classification accuracy of over 83%, which is better than the state-of-the-art models, while maintaining high interpretability.

## 1. Introduction

Coronavirus disease 2019 (COVID-19) is caused by severe acute respiratory syndrome coronavirus (SARS-CoV-2), which has rapidly spread worldwide and posed significant public health threats. As of 25 January 2021, there have been more than 99.257 million confirmed cases of COVID-19 and 2,131,000 cumulative deaths worldwide, according to the Global COVID-19 Status Report released by Johns Hopkins University (https://coronavirus.jhu.edu/map.html (accessed on 1 August 2021)). The number of confirmed/fatal cases is still on the rise. Currently, the main screening methods for COVID-19 cases include reverse transcription polymerase chain reaction (RT-PCR) or gene sequencing for respiratory or blood specimens [1]. However, the overall RT-PCR positive rate of throat swab samples is reported to be 30 to 60%; thus, subjects with false-negative test results may continue to transmit the virus within a community [2]. CXR is a routine diagnostic tool incorporated in the initial diagnosis and subsequent monitoring for COVID-19 patients. It boasts short examination time, low radiation dose and low cost of examination. CXR images show visual indexes correlated with COVID-19 [3]. Reported CXR features include airspace opacities, which can show multilobar involvement, peripheral dominance, asymmetric or diffuse involvement [4]. Early changes on CXR are often subtle, requiring radiologist expertise. Nevertheless, radiology expertise is scarce in developing countries, which has been further strained by the tremendous case load of COVID-19 infection. Applying artificial intelligence (AI)/machine learning approaches for the automated identification of subtle CXR abnormalities serves as a potential remedy [5,6].

At present, several AI models have been developed for COVID-19 detection using chest radiography images [7,8,9]. Minaee et al. used transfer learning to train four famous convolutional neural networks (ResNet18, ResNet50, SqueezeNet and DenseNet-121) for identifying COVID-19 disease in the analyzed CXR images [10]. The results showed that although most of these networks achieved a sensitivity rate of 98% (±3%), only around a 90% specificity rate was achieved. Ozturk et al. proposed a deep learning model for detecting and classifying COVID-19 cases using CXR images [11]. The proposed model was fully automated with an end-to-end structure without the need for manual feature extraction and produced a classification accuracy of 87.02% for multi-class prediction (COVID-19 vs. No-Findings vs. Pneumonia). Table 1 summarizes part of the recent studies on AI model development for COVID-19 detection using chest radiography images. It can be observed that deep learning models are in great favor due to the promising performance; research in this area is anticipated to grow. In spite of this, almost all deep learning models have been regarded as a black box given the incomprehensible deep features and lack of transparency in the training procedures, raising a concern regarding the model interpretability. Of note, when it comes to adopting the AI models as clinical decision support systems, model interpretability is an indispensable prerequisite [12].

In this regard, Takagi–Sugeno–Kang (TSK) fuzzy systems are also in great favor owing to their capability to maintain outstanding balance between approximation ability and interpretability [24,25,26]. Although fuzzy systems were originally designed for regression or binary tasks (it can be considered as special regression tasks), by virtue of strategic approaches, fuzzy systems have achieved satisfactory performance in clinical diagnosis, mostly in disease classification. For example, Jiang et al. proposed a multi-view TSK fuzzy system for epileptic EEG signal detection [27]. Hsieh et al. combined an adaptive neuro-fuzzy inference system (ANFIS) with greedy forward feature selection to develop an intelligent diagnostic system for differentiating cases with influenza from those with cold [28]. Khayamnia et al. established a Mamdani fuzzy system for migraine headache diagnosis [29]. In each of the above studies, the label vector was transformed into a strict binary matrix in order to apply the fuzzy system for multi-class classification tasks. However, numerous studies indicated that transforming the training samples to a strict binary label matrix is too rigid to learn a discriminative transformation matrix [30,31,32].

Confronted with all these, in this study, we developed a TSK fussy system to enhance the interpretability of our AI model for COVID-19 detection (COVID-19 vs. No-Findings vs. Pneumonia). Meanwhile, we introduced a soft strategy to adaptively relax the binary matrix during the label transformation. Inspired by manifold learning, a class compactness graph was constructed to minimize the risk of model overfitting following the introduction of label relaxation. The core idea lies in that samples sharing the same labels should be kept as close as possible when they are transformed into the label space. The contributions of this study can be summarized from the model and experiment perspectives.

(1)When classic TSK fuzzy systems are applied to classification tasks, margins between different classes are expected to be as large as possible after they are transformed into their label space. However, this assumption is too rigid to learn a discriminative transformation matrix. Additionally, excessive label fitting may cause overfitting. To address these issues, label softening and the compactness class graph are embedded into the objective function of the classic TSK fuzzy systems, which can bring two advantages: firstly, the new method can enlarge the margins between different classes as much as possible, and it has more freedom to fit the labels better. Secondly, to avoid the problem of overfitting, the new method uses the class compactness graph to guarantee that the samples from the same class can be kept close together in the transformed space.(2)Five state-of-the-art algorithms are introduced for comparison studies, and experimental results are reported from the perspectives of classification performance, sensitivity and interpretability. The proposed algorithm significantly performs better than the state-of-the-art algorithm in terms of accuracy and macro F1-score due to the introduced label soften and the compactness class graph. What is more, the improved generalization ability can be observed from the lower accuracy difference between training and testing. In addition, radiomics features have physical meaning; with the help of transparent fuzzy rules generated by the proposed algorithm, the interpretability can be guaranteed.

The rest of the sections are organized as follows: In Section 2, we briefly describe the working mechanism of the zero-order TSK fuzzy system for classification tasks. In Section 3, we introduce the objective function, optimization, and the corresponding algorithm of our proposed TSK fuzzy system. In Section 4, we conduct experiments on radiomics features extracted from CXR images for model evaluation. In Section 5, we discuss the experimental results and conclude our study. The abbreviations and symbols used in this study are given in the Abbreviation section. 

## 2. ZERO-TSK-FS for Classification

The original TSK fuzzy systems were designed for regression tasks. However, the diagnosis of COVID-19 based on radiomics belongs to a classification task, as we know that the zero-order TSK fuzzy system (ZERO-TSK-FS) has very concise fuzzy rules and is widely used due to low model complexity. Therefore, in this section, we re-design ZERO-TSK-FS specifically for the classification task. Suppose we have a training set $\chi ={\left\{(x_{i},y_{i})\right\}}_{i=1}^{N}$; each input $x_{i}$ is represented in the *d*-dimensional feature space ${\left[x_{i1},x_{i2},\ldots ,x_{id}\right]}^{T}\in R^{d}$, and $y_{i}\in R$ is the corresponding label. Then, the *k*-th fuzzy rule of ZERO-TSK-FS can be defined as follows,
(1)$$\mathrm{If}\;x_{i1}\;\mathrm{is}\;\vartheta_{1}^{k}\wedge x_{i2}\;\mathrm{is}\;\vartheta_{2}^{k}\wedge \ldots \wedge x_{id}\;\mathrm{is}\;\vartheta_{d}^{k},\;\mathrm{then}\;f^{k}(x_{i})=\beta^{k},k=1,2,\ldots ,K,$$
where $\vartheta_{j}^{k}$ represents a fuzzy set subscribed by feature $x_{ij}$ for the *k*-th rule, ∧ is a fuzzy conjunction operator, and *K* is the number of fuzzy rules. Each fuzzy rule is premised on the input feature space ${\left[x_{i1},x_{i2},\ldots ,x_{id}\right]}^{T}\in R^{d}$ and transforms the fuzzy sets in the input feature space into a varying singleton represented by $f^{k}(x_{i})$. After a series of operation and defuzzification steps, we can obtain the output of $x_{i}$, i.e., $f(x_{i})$,
(2)$$f(x_{i})=\sum_{k=1}^{K}f^{k}(x_{i})\varphi^{k}(x_{i},\mu^{k},\delta^{k})=\sum_{k=1}^{K}\beta^{k}\varphi^{k}(x_{i},\mu^{k},\delta^{k}),$$
where the normalized fuzzy membership function $\varphi^{k}(x_{i},\mu^{k},\delta^{k})$ of the *k*-th fuzzy rule can be formulated as
(3)$$\varphi^{k}(x_{i},\mu^{k},\delta^{k})=\frac{{\tilde{\varphi }}^{k}(x_{i},\mu^{k},\delta^{k})}{\sum_{k\prime =1}^{K}{\tilde{\varphi }}^{k\prime }(x_{i},\mu^{k\prime },\delta^{k\prime })}.$$

In (3), ${\tilde{\varphi }}^{k}(x_{i},\mu^{k},\delta^{k})$ can be computed by $\prod_{j=1}^{d}{\tilde{\varphi }}_{\vartheta_{d}^{k}}^{k}(x_{ij})$. If the Gaussian fuzzy membership function is adopted, we have
(4)$${\tilde{\varphi }}^{k}(x_{i},\mu^{k},\delta^{k})=\prod_{j=1}^{d}\exp(\frac{-(x_{ij}-\mu_{j}^{k})}{2\delta_{j}^{k}}),$$
where $\mu^{k}=[\mu_{1}^{k},\mu_{2}^{k},\ldots ,\mu_{d}^{k}]$ and $\delta^{k}=[\delta_{1}^{k},\delta_{2}^{k},\ldots ,\delta_{d}^{k}]$ denote the kernel center vector and kernel width vector, respectively. By denoting $\phi (x_{i})=[\varphi^{1}(x_{i},\mu^{1},\delta^{1}),\varphi^{2}(x_{i},\mu^{2},\delta^{2}),\ldots ,\varphi^{K}(x_{i},\mu^{K},\delta^{K})]$, and $\beta =[\beta^{1},\beta^{2},\ldots ,\beta^{K}{]}^{T}$, then $f(x_{i})$ in (3) can be updated as
(5)$$f(x_{i})=\phi (x_{i})\beta .$$

It can be seen that the optimization of the consequent β can be actually considered as solving a linear regression problem in (5). There are many algorithms that can be used to solve this problem. A representative one proposed by Deng et al. has been demonstrated its promising performance [33], which can be formulated as
(6)$$\underset{\beta }{\min}{\left\|\Theta \beta -y\right\|}_{F}^{2}+\lambda {\left\|\beta \right\|}_{F}^{2}.$$
where $\Theta =[\phi (x_{1}),\phi (x_{2}),\ldots ,\phi (x_{N}){]}^{T}$, and $y=[y_{1},y_{2},\ldots ,y_{N}{]}^{T}\in R^{N\times 1}$ is the output vector. Notably, the model obtained from (6) is a regression model that cannot be directly used for our classification task. Usually, when we use a regression model for a multi-class classification task, a label transformation should be conducted [27]. That is to say, we should transform the label space from $R^{N\times 1}$ to $R^{N\times C}$, where *C* is the number of classes. For example, suppose we have a three-class classification training set $\chi ={\left\{(x_{i},y_{i})\right\}}_{i=1}^{N}$ having four samples, where $y=[2,1,3,1{]}^{T}\in R^{4\times 1}$; then, after label space transformation, we have the new label space $Y\in R^{4\times 3}$,
(7)$$Y=\left[\begin{array}{ccc} 0 & 1 & 0\\ 1 & 0 & 0\\ 0 & 0 & 1\\ 1 & 0 & 0 \end{array}\right],$$
which means that the four samples belong to the second, first, third and first class, respectively. Therefore, by substituting the new label space $Y\in R^{N\times C}$ into (6) to replace the original label space $y=R^{N\times 1}$, we have
(8)$$\underset{\Delta }{\min}{\left\|\Theta \Delta -Y\right\|}_{F}^{2}+\lambda {\left\|\Delta \right\|}_{F}^{2}.$$

Here, we use $\Delta \in R^{K\times C}$ as the consequent matrix to replace $\beta \in R^{K\times 1}$ in (6). In this manner of label transformation, ZERO-TSK-FS can be used for the multi-class classification task by transforming the training samples to a strict binary label matrix, such as $Y\in R^{N\times C}$ shown in (7).

However, as we previously stated, transforming the training samples to a strict binary label matrix may not be adequately flexible to learn a discriminative transformation matrix. Therefore, in the following sections, we develop a novel zero TSK fuzzy system for COVID-19 cases detection which can relax the strict binary label matrix to a soft one.

## 3. LR-ZERO-TSK-FS

### 3.1. Objective Function

To fit the labels in a soft way, we introduced a non-negative label soften matrix $\Omega \in R^{N\times C}$ and a luxury matrix $\Xi \in R^{N\times C}$ to relax the strict binary label matrix. Let us also take the three-class classification training set having four samples as an example; with Ω and Ξ, the strict binary label matrix can be relaxed into the following form:(9)$$\tilde{Y}=Y+\Omega ⊙\Xi =\left[\begin{array}{ccc} -\Omega_{11} & 1+\Omega_{12} & -\Omega_{13}\\ 1+\Omega_{21} & -\Omega_{22} & -\Omega_{23}\\ -\Omega_{31} & -\Omega_{32} & 1+\Omega_{33}\\ 1+\Omega_{41} & -\Omega_{42} & -\Omega_{43} \end{array}\right],$$
where the non-negative label matrix $\Omega \in R^{N\times C}$ is defined as
(10)$$\begin{array}{l} \Omega =\left[\begin{array}{ccc} \Omega_{11} & \ldots & \Omega_{1C}\\ \Omega_{21} & \ldots & \Omega_{2C}\\ \ldots & \ldots & \ldots \\ \Omega_{N1} & \ldots & \Omega_{NC} \end{array}\right]\\ s.t.\;\Omega_{ij}\geq 0, \end{array}$$
the luxury matrix $\Xi \in R^{N\times C}$ is defined as
(11)$$\Xi (i,j)=\left\{\begin{array}{c} +1\;\mathrm{if}\;Y(i,j)=1\\ -1\;\mathrm{if}\;Y(i,j)=0 \end{array}\right.,$$
and ⊙ is a Hadamard product operator of matrices.

Therefore, with label matrix relaxation, the objective function of ZERO-TSK-FS in (8) can be updated as
(12)$$\underset{\Delta }{\min}{\left\|\Theta \Delta -(Y+\Omega ⊙\Xi )\right\|}_{F}^{2}+\lambda {\left\|\Delta \right\|}_{F}^{2}.$$

Although label relaxation can help learn a more discriminant transformation matrix Δ, it may lead to overfitting problems. In manifold learning, the class compactness graph is often used to alleviate overfitting problems. The core idea is that samples sharing the same labels should be kept as close as possible when they are transformed into the label space. More specifically, a weight matrix W is defined as follows to capture the relationship that the samples sharing the same labels should be kept as close as possible,
(13)$$W(i,j)=\left\{\begin{array}{l} e^{-{\left\|x_{i}-x_{j}\right\|}^{2}/\delta \;}if\;x_{i}\;and\;x_{j}\;\mathrm{have}\;\mathrm{the}\;\mathrm{same}\;\mathrm{labels}\;\\ 0\;\;\;\;\;\;\;\;\;\;\;\;\;\;\mathrm{otherwise} \end{array}\right..$$

In the class compactness graph, if sample $x_{i}$ and sample $x_{j}$ have the same labels, they are connected by an undirected edge, and the corresponding weight can be computed by (13). We find that the closer sample $x_{i}$ and sample $x_{j}$ are, the bigger the corresponding weight. Therefore, when all training samples in the training space are transformed into the label space, any two samples having bigger weights in the training space should also be kept as close as possible in the label space. That is to say, it is reasonable to minimize the following objective to achieve our goal,
(14)$$\underset{f}{\min}\sum_{i=1}^{N}\sum_{j=1}^{N}{\left\|f(x_{i})-f(x_{j})\right\|}^{2}W_{ij},$$
where $W_{ij}$ is the *i*-th row and *j*-th column element in the weight matrix W, and $f(x_{i})=\phi (x_{i})\Delta$ is the transformation result of $x_{i}$ represented in the label space. By substituting $f(x_{i})=\phi (x_{i})\Delta$ to (14) and performing equivalent mathematical transformation, we have
(15)$$\underset{\Delta }{\min}\mathrm{tr}(\Delta^{T}\Theta^{T}L\Theta \Delta ),$$
where L is the graph Laplacian, which is defined as L=Z−W. Z is a diagonal matrix, and the diagonal element can be computed by $Z_{ij}=\sum_{j=1}^{N}W_{ij}$.

Finally, by embedding (15) to (12), we have the following objective function for LR-ZERO-TSK-FS,
(16)$$\underset{\Delta ,\Omega }{\min}{\left\|\Theta \Delta -(Y+\Omega ⊙\Xi )\right\|}_{F}^{2}+\lambda {\left\|\Delta \right\|}_{F}^{2}+\gamma \mathrm{tr}(\Delta^{T}\Theta^{T}L\Theta \Delta ).$$

### 3.2. Optimization

The objective function shown in (16) has two components that need to be optimized, i.e., the transformation matrix Δ and the non-negative label relaxation matrix Ω. Thus, we introduced an iterative strategy to search for the optimal solution for Δ and Ω [34].

Firstly, suppose that the non-negative label relaxation matrix Ω is fixed; then, the optimization problem becomes
(17)$$J(A)=\underset{\Delta }{\min}{\left\|\Theta \Delta -U\right\|}_{F}^{2}+\lambda {\left\|\Delta \right\|}_{F}^{2}+\gamma \mathrm{tr}(\Delta^{T}\Theta^{T}L\Theta \Delta ),$$
where U=Y+Ω⊙Ξ. By setting the partial derivative with respect to Δ to 0, we have
(18)$$\begin{array}{l} \partial J(A)/\partial \Delta =2\Theta^{T}\Theta \Delta -2\Theta^{T}U+2\lambda \Delta +2\gamma \Theta^{T}L\Theta \Delta =0\\ \Rightarrow \Delta ={\left(\Theta^{T}\Theta +\lambda I+\gamma \Theta^{T}L\Theta \right)}^{-1}\Theta^{T}U\\ ={\left(\Theta^{T}\Theta +\lambda I+\gamma \Theta^{T}L\Theta \right)}^{-1}\Theta^{T}(Y+\Omega ⊙\Xi ) \end{array}$$

Secondly, suppose that the transformation matrix Δ is fixed; since the second term in (16) is uncorrelated with Ω, the optimization problem becomes
(19)$$\begin{array}{l} J(\Omega )=\underset{\Omega }{\min}{\left\|V-\Omega ⊙\Xi \right\|}_{F}^{2}\\ s.t.\;\Omega \geq 0 \end{array}$$
where V=ΘΔ−Y. We have common knowledge that the squared Frobenius norm of the matrix can be decoupled element by element. Therefore, the optimization problem in (19) can be equivalently decoupled into N×C sub-problems. For the *i*-th row and *j*-th column element $\Omega_{ij}$ in Ω, we have the corresponding sub-problem,
(20)$$\begin{array}{l} J(\Omega_{ij})=\underset{\Omega_{ij}}{\min}{\left\|V_{ij}-\Omega_{ij}\Xi_{ij}\right\|}_{F}^{2}\\ s.t.\;\Omega_{ij}\geq 0 \end{array}$$

Given that ${\left(\Xi_{ij}\right)}^{2}=1$, we have ${\left(V_{ij}-\Omega_{ij}\Xi_{ij}\right)}^{2}={\left(\Xi_{ij}V_{ij}-\Omega_{ij}\right)}^{2}$. Considering the non-negative constraint imposed on $\Omega_{ij}$, we can obtain $\Omega_{ij}=\max(\Xi_{ij}V_{ij},0)$. Accordingly, Ω in (19) can be computed as
(21)Ω=max(Ξ⊙V,0).

With the closed-form solution of Δ in (18) and Ω in (21), we can use an iterative strategy to find their optimal values.

### 3.3. Algorithm

The training steps of LR-ZERO-TSK-FS are shown in Algorithm 1. We can see that the asymptotic time complexity of LR-ZERO-TSK-FS is mainly contributed by two components, i.e., FCM clustering for antecedent learning and the computing of Δ for consequent learning. The time complexity of FCM clustering is $O(d^{2}NK)$, where d is the dimension of the training set. The time complexity of the computing of Δ is $O(K^{3})$. Therefore, the asymptotic time complexity of LR-ZERO-TSK-FS is $O(d^{2}NK+K^{3})$. Since d and K are relatively small compared with *N*, the asymptotic time complexity of LR-ZERO-TSK-FS can be considered linear with *N*.

**Algorithm 1** LR-ZERO-TSK-FS
**Input**: Training set $\chi ={\left\{(x_{i},y_{i})\right\}}_{i=1}^{N}$, number of fuzzy rules *K*, regularized parameters λ and γ

**Output**: Transformation matrix Δ and non-negative label soften matrix Ω

**Procedures**:

**^1^** Use clustering technique, e.g., FCM (Fuzzy C-Mean) to learn the antecedent parameters, i.e., $\mu_{j}^{k}$ and $\delta_{j}^{k}$ in (4) of fuzzy rules.

**^2^** Use (3) to get $\phi (x_{i})$, and further Θ.

**^3^** Use (13) and $Z_{ij}=\sum_{j=1}^{N}W_{ij}$ to compute the graph Laplacian matrix L.

**^4^** Randomize Ω under the constraint Ω≥0.

**^5^** Set t←0.


**Repeat**

  **  ^6^** Update Δ(t+1) by (19) with current Ω(t).

  **  ^7^** Update Ω(t+1) by (21) with current Δ(t+1).

  **  ^8^**
t←t+1.

**Until** ${\left\|\Delta (t+1)-\Delta (t)\right\|}^{2}\leq \varepsilon$
**^9^** With Δ and Ω, the output can be computed by Y=ΘΔ−(Ω⊙Ξ).



## 4. Experimental Studies

### 4.1. Data Preprocessing

We collected 600 chest X-Ray (CXR) images from 200 normal cases, 200 COVID-19 cases, and 200 non-COVID-19 pneumonia cases, which are publicly available from online databases (https://github.com/agchung/Figure1-COVID-chestxray-dataset (accessed on 8 August 2021); https://github.com/agchung/Actualmed-COVID-chestxray-dataset (accessed on 8 August 2021); https://www.kaggle.com/tawsifurrahman/covid19-radiography-database (accessed on 8 August 2021); https://www.kaggle.com/c/rsna-pneumonia-detection-challenge/data (accessed on 8 August 2021)). The workflow of data preprocessing is shown in Figure 1.

In data preprocessing, we aimed to extract radiomics features from CXR images. Firstly, a binary lung region mask was generated by a pretrained U-NET segmentation network. With the obtained lung region on each CXR image, we then used the radiomics package to extract different types of radiomics features for downstream modeling.

### 4.2. Settings

After extracting radiomics features from CXR images, we followed the workflow, as shown in Figure 2, to evaluate the proposed model.

To simulate an independent test, we introduce holdout cross-validation, as shown in Figure 2. To be specific, we firstly independently select a holdout set as a testing set. Then, the rest is cut into *K* (*K* = 3 in this study) folds; one is taken as the validation set and the rest are taken as the training set. In the validation phase, minimum-redundancy-maximum-relevance (mRMR) is adopted as a feature-ranking algorithm. Then, the cross-validation (3-CV) strategy was used to determine the optimal feature set and hyper-parameters with respect to the proposed model. In the training phase, with the optimal feature set and hyper-parameters, the best model can be obtained. In the testing phase, with the best model, we can obtain the corresponding testing results.

To highlight the performance of the proposed model, we introduced several benchmarking models for comparison. They are ZERO-TSK-FS [35], L2-TSK-FS [24], FS-FCSVM [36], linear SVM (L-SVM) and Gaussian SVM (G-SVM) [37], respectively. Table 2 gives the experimental settings.

The criteria *accuracy* and *macro F1-score* were introduced to quantitatively evaluate the classification performance of all studied models.

### 4.3. Experimental Results

In this section, we report our evaluation results from three main aspects, i.e., classification performance analysis, sensitivity analysis and interpretability analysis.

#### 4.3.1. Classification Performance Analysis

The testing and training accuracy of the proposed model LR-ZERO-TSK-FS and the studied benchmarking models are shown in Figure 3. The horizontal axis represents the top-*k* features selected by using the feature selection method, as shown in Figure 2. It can be observed that LR-ZERO-TSK-FS wins the best performance under the most top-*k* features except top-5 and top-35. Specially, LR-ZERO-TSK-FS always performs better than the baseline ZERO-TSK-FS on the testing set. As indicated in Figure 3b, LR-ZERO-TSK-FS also wins the best from the top-20 to top-50 features. Statistical analysis of the testing performance in terms of *t*-test is shown in Table 3, where “*” means that there exist significant differences between LR-ZERO-TSK-FS and the state-of-the-art models (*p* < 0.05). A similar observation can also be obtained by using another metric macro F1-score, as shown in Table 4 and Table 5. In Table 6, we also report the CPU seconds each algorithm consumed.

To further investigate the impact of training sample size on the difference in model accuracy between training and testing sets (i.e., model generalizability), we trained the models under varying training sample sizes. Figure 4 shows the accuracy differences (the absolute error of training accuracy and test accuracy) between the training and testing datasets, where the horizontal axis represents the size of the training samples. We have common knowledge that fewer training samples are more likely to result in model overfitting, which is also demonstrated in Figure 4. It can also be noticed that the proposed model LR-ZERO-TSK-FS is more effective in inhibiting model overfitting compared with the benchmarking models. As we stated in the introduction part, label relaxation can help to learn a more discriminant transformation matrix, but it also tends to result in overfitting. This statement can be demonstrated by the results shown in Figure 5.

When the hyperparameter γ is set to 0, the manifold regularization term in (15) becomes useless. From Figure 5a, it can be observed that LR-ZERO-TSK-FS with γ=0 has larger accuracy differences than the benchmarking models. When compared with ZERO-TSK-FS, in particular, the result demonstrated that the label relaxation indeed aggravates overfitting in the absence of manifold regularization. When the manifold regularization term was activated, see Figure 5b, the advantageous testing accuracy can basically be maintained. That is to say, the combination of label relaxation and manifold regularization achieve an outstanding balance between classification accuracy and generalization.

#### 4.3.2. Sensitivity Analysis

The proposed model LR-ZERO-TSK-FS has two regularization hyper-parameters λ and γ that needed to be set in advance. In this study, we analyzed the robustness of LR-ZERO-TSK-FS with respect to λ and γ. As illustrated in Figure 6, it can be observed that LR-ZERO-TSK-FS is sensitive to λ and γ, and smaller λ and γ seem to yield better testing accuracy. Therefore, according to Figure 6, λ and γ can be determined by using cross-validation under a small range.

Results from comparative analyses indicated that LR-ZERO-TSK-FS yielded the best overall model performance on the COVID-19 cases detection task (Figure 3). From Figure 5b, we can notice that when the manifold regularization is not activated (γ=0), LR-ZERO-TSK-FS also performs satisfactorily and better than LR-ZERO-TSK-FS with manifold regularization. That is to say, from Figure 3 and Figure 5b, the capability of our label relaxation strategy for enhancing the classification performance of the TSK fuzzy systems was successfully demonstrated. Moreover, from Figure 4 and Figure 5a, we can see that when the manifold regularization is activated, the differences in accuracy between the training and testing sets remarkably drop. In particular, the comparative analysis with the ZERO-TSK-FS explicitly indicated the superiority of LR-ZERO-TSK-FS in minimizing the risk of model overfitting. It is noteworthy that the present evidence relies on the following particularities of LR-ZERO-TSK-FS:(1)Streamlined classification accuracy owing to the introduction of a soft strategy to adaptively relax the binary matrix during label transformation, rendering more flexibility during label transformation and capability in enlarging the margins between different classes.(2)Alleviated risk of model overfitting in virtue of the adoption of a class compactness graph during manifold learning, based on the assumption that samples sharing the same labels should be kept as close as possible when they are transformed into the label space.

#### 4.3.3. Interpretability Analysis

The proposed model LR-ZERO-TSK-FS was derived from the zero order TSK fuzzy system, carrying the inherent characteristic of model interpretability. Therefore, in the following, we give an example to illustrate how the proposed model can diagnose a subject with COVID-19 with the use of radiomics features.

The optimal radiomics features selected by step 3 shown in Figure 2 are listed in Table 7, which were used for model training. The training results (five trained fuzzy rules) of LR-ZERO-TSK-FS using the selected radiomics features in terms of antecedent and consequent parameters are listed in Table 8.

Figure 7 shows the fuzzy membership function of one radiomics feature (No. 1: wavelet-HH_glrlm_GrayLevelVariance) trained on five fuzzy rules. According to the center $\mu^{k}$ of each fuzzy membership function and domain knowledge, we can assign a linguistic description, e.g., “low”, “a little low”, “medium”, “a little high” and “high”, to each feature. For example, from the fuzzy mapping of “wavelet-HH_glrlm_GrayLevelVariance” in the first fuzzy rule, it can be interpreted that “wavelet-HH_glrlm_GrayLevelVariance” is “low”. According to Yang et al. [38], linguistic meanings are dependent on expert knowledge. This is the beauty of interpretable models because the acquired knowledge by models can be refined and integrated with expert knowledge. Accordingly, the trained five fuzzy rules are described in Table 9.

Under these five trained fuzzy rules, COVID-19 detection of an individual can be made following the procedures illustrated in Figure 8.

From Figure 8, it can be found that the classification of the case depends on the obtained five fuzzy rules. As shown in Table 6, according to the parameters learned in the if-part, the five fuzzy rules can be assigned different linguistic meaning based on domain knowledge. Therefore, each fuzzy rule actually indicates the classification result of one expert (or one kind of knowledge). For example, the first fuzzy rule classifies the case to COVID-19 based on the domain knowledge embedded in the if-part. Finally, by linearly combining all fuzzy rules (combing knowledge of all experts), the case is classified to COVID-19.

## 5. Conclusions

COVID-19 has posed a significant public health threat globally. The automated detection of CXR abnormality potentially aids identifying patients with significant risk of COVID infection earlier. In this study, we developed an interpretable and soft-label-driven TSK fuzzy system for multi-class COVID-19 detection (COVID-19 vs. No-Findings vs. Pneumonia) using radiomics features extracted from CXR images. The risk of model overfitting is alleviated in virtue of the adoption of a class compactness graph during manifold learning, which is based on the assumption that samples sharing the same labels should be kept as close as possible when they are transformed into the label space. We successfully demonstrated that our model outperformed the comparable state-of-the-art models while maintaining high interpretability.

This study is not without limitations. For example, in the construction of the class compactness graph, we just used Euclidean distance. In addition, in our experiments, multi-center based external validation is not carried out. Therefore, in our future work, we will carry out more in-depth research from the above aspects.

## Figures and Tables

**Figure 1 diagnostics-12-02613-f001:**
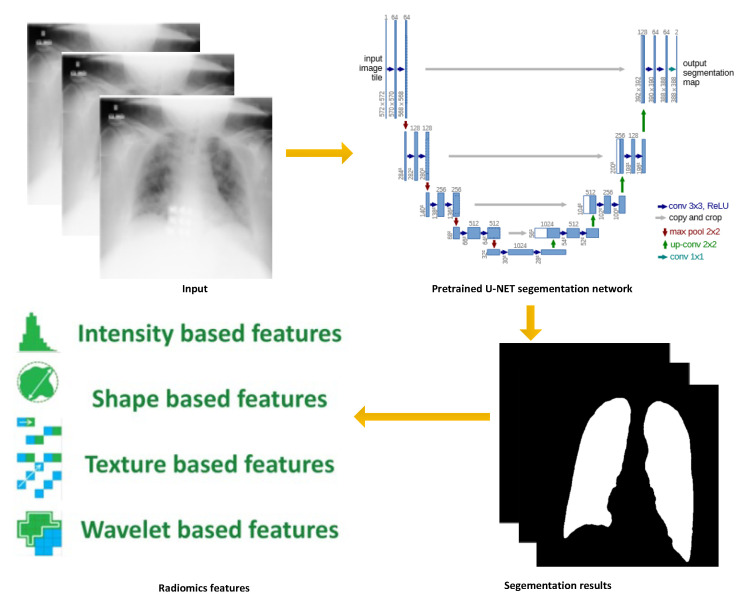
Workflow of data preprocessing.

**Figure 2 diagnostics-12-02613-f002:**
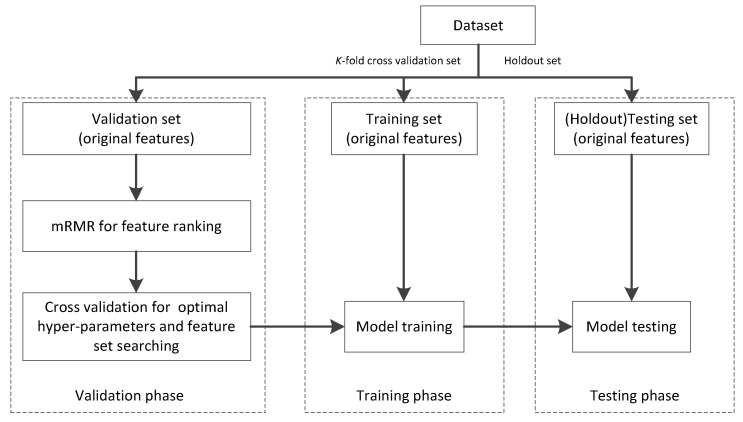
Workflow of model evaluation.

**Figure 3 diagnostics-12-02613-f003:**
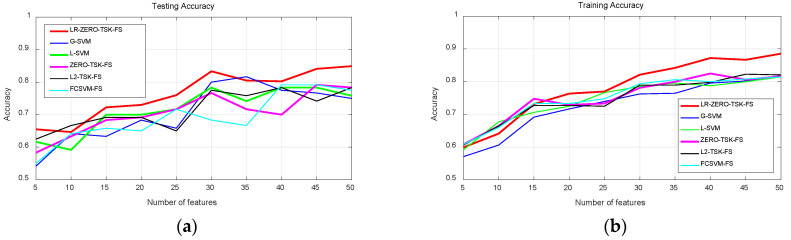
Classification performance in terms of accuracy of all models. (**a**) Testing accuracy. (**b**) Training accuracy.

**Figure 4 diagnostics-12-02613-f004:**
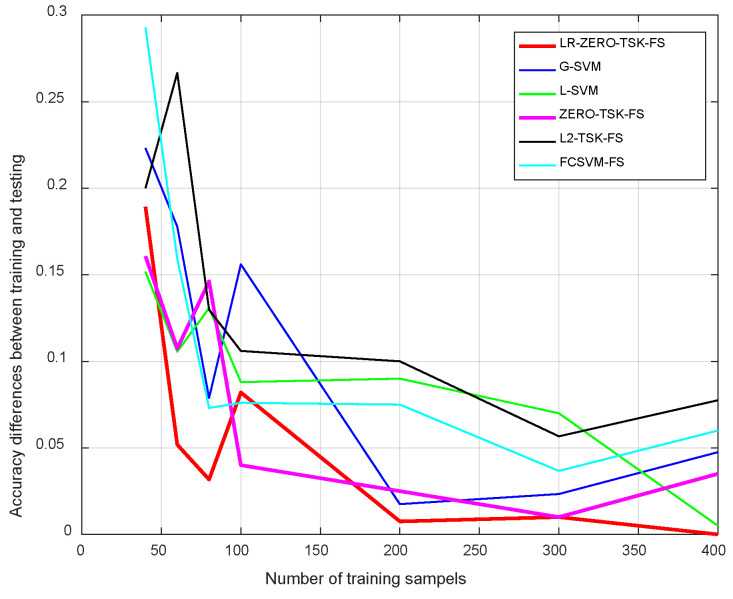
Accuracy differences between training and testing sets.

**Figure 5 diagnostics-12-02613-f005:**
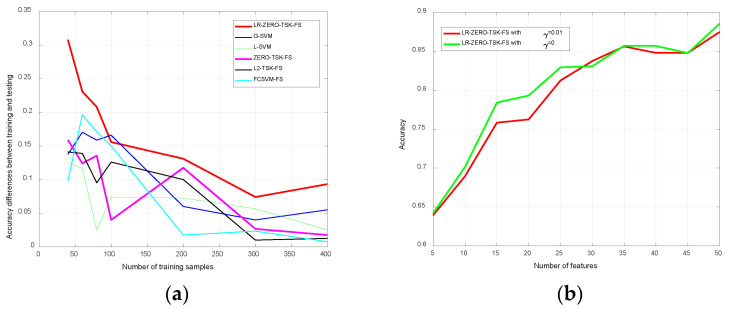
Evaluation of manifold regularization. (**a**) Accuracy difference comparison when manifold regularization is absent. (**b**) Accuracy comparison between γ=0 and γ=0.01.

**Figure 6 diagnostics-12-02613-f006:**
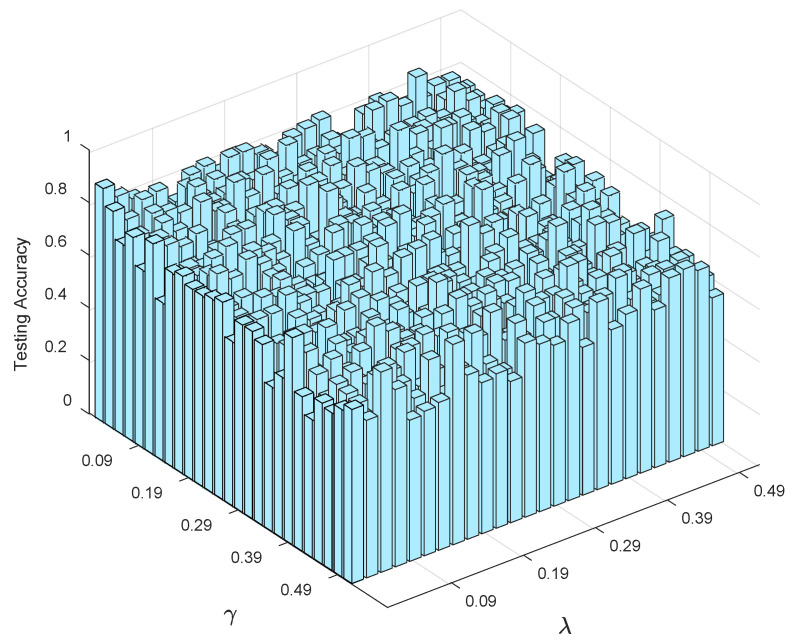
Robustness analysis with respect to λ and γ.

**Figure 7 diagnostics-12-02613-f007:**
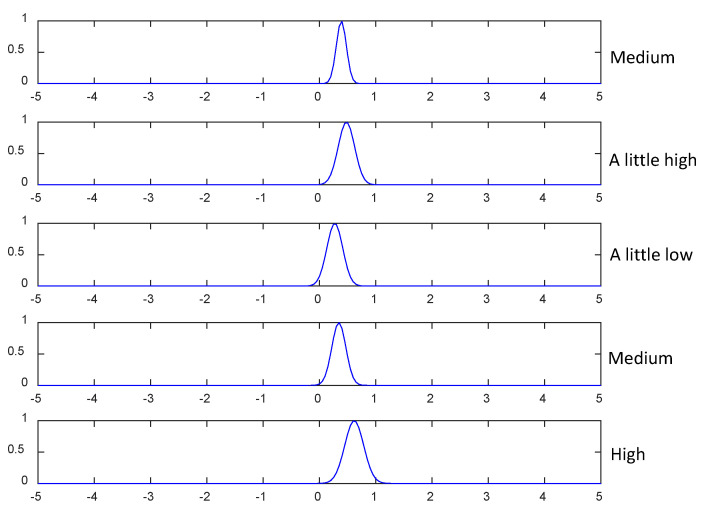
The fuzzy membership function and the corresponding linguistic description of “wavelet-HH_glrlm_GrayLevel Variance” in each fuzzy rule.

**Figure 8 diagnostics-12-02613-f008:**
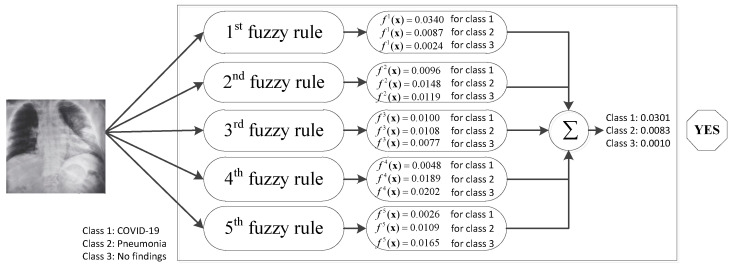
Interpretable diagnostic procedure.

**Table 1 diagnostics-12-02613-t001:** AI models for COVID-19 cases detection on chest radiography images.

Studies	Modalities	Number of Cases	AI Models	Best Performance
COVID-19 detection from healthy and pneumonia cases [13].	CCT *	219 COVID-19 cases (S) 224 Pneumonia cases 175 Healthy cases	ResNet with location attention mechanism	Overall accuracy rate was 86.7%.
Severity rating [14].	CCT	313 COVID-19 cases (S) 229 COVID-19 cases (M)	UNet and 3D Deep Network	The model obtained a testing AUC of 0.975.
Severity rating [15].	CCT	195 COVID-19 cases (S) 258 COVID-19 cases (M)	M-Inception	The external testing dataset showed a total accuracy of 79.3% with a specificity of 0.83 and sensitivity of 0.67.
COVID-19 detection from healthy cases [16].	CCT	777 COVID-19 cases (S) 708 Healthy cases	DRE-Net	The model discriminated the COVID-19 patients from the bacteria pneumonia patients with an AUC of 0.95, recall (sensitivity) of 0.96, and precision of 0.79.
Severity rating [17].	CXR **	50 COVID-19 cases (S) 50 COVID-19 cases (M)	Deep CNN and ResNet-50	The model achieved 99.7% accuracy for automatic detection of COVID-19.
COVID-19 detection from healthy cases [18].	CXR	25 COVID-19 cases (S) 25 Healthy cases	COVIDX-Net	The model achieved f1-scores of 0.89 and 0.91 for normal and COVID-19, respectively.
COVID-19 detection from healthy cases [19].	CXR	25 COVID-19 cases (S) 25 COVID-19 cases (M)	ResNet50 and SVM	The highest accuracy achieved by ResNet50 plus SVM is 98.66%.
COVID-19 detection from healthy cases [20].	CXR	53 COVID-19 cases (S) 5526 COVID-19 cases (M)	COVID-Net	COVID-Net achieves good accuracy by achieving 93.3% test accuracy.
COVID-19 detection from healthy and pneumonia cases [21].	CXR	224 COVID-19 cases (S) 700 Pneumonia cases 504 Healthy cases	VGG-19	The best accuracy, sensitivity, and specificity obtained is 96.78%, 98.66%, and 96.46% respectively.
COVID-19 detection from healthy and pneumonia cases [22]	CXR	15,959 CXR images of 15,854 patients, covering normal, pneumonia, and COVID-19 cases.	Deep CNN	The model achieved best positive predictive value of 91.6%, 92.45%, and 96.12%, respectively for normal, pneumonia, and COVID-19 cases, respectively.
COVID-19 detection from healthy and pneumonia cases [23]	CXR	250 COVID-19 cases 2753 pulmonary cases 3520 healthy cases.	Deep CNN	The best accuracy is 0.99.

*: Chest CT; **: Chest X-Ray. M: mild. S: severity.

**Table 2 diagnostics-12-02613-t002:** Experimental settings.

Model	Parameter Settings
ZERO-TSK-FS	FCM is used to learn the antecedent parameters. The optimal number of fuzzy rules is determined by cross-validation from [2, 4, 6, …, 30].
L2-TSK-FS	FCM is used to learn the antecedent parameters. The optimal number of fuzzy rules is determined by cross-validation from [2, 4, 6, …, 30].
FS-FCSVM	FCM is used to learn the antecedent parameters. The optimal number of fuzzy rules is determined by cross-validation from [2, 4, 6, …, 30]. The learning threshold parameter is determined by cross-validation from [0.2, 0.3, …, 0.8]. The regularization parameter is determined by cross-validation from [2^−3^, 2^−2^, …, 2^5^, 2^7^].
L-SVM	The regularization parameter *C* is determined by cross-validation from [10^−3^, 2^−2^, …, 10^3^].
G-SVM	The regularization parameter *C* is determined by cross-validation from [10^−3^, 2^−2^, …, 10^3^]. The kernel width σ is determined by cross-validation from [10^−3^, 10^−2^, …, 10^3^].
LR-ZERO-TSK-FS	FCM is used to learn the antecedent parameters. The optimal number of fuzzy rules is determined by cross-validation from [2, 4, 6, …, 30]. The regularization parameter λ and γ are both determined by cross-validation from [0.01, 0.03, …, 1].
Software and hardware settings	
Development platform: Python 3.9.8 (pyRadiomics for radiomics feature extraction), Matlab 2012b (LR-ZERO-TSK-FS coding) System OS: Windows 10 Hardware: Intel(R) Core (TM) i5-7200U CPU @ 2.50 GHz 2.71 GHz, 16 G RAM	

**Table 3 diagnostics-12-02613-t003:** Statistical analysis of testing performance in terms of *t*-test.

Number of Features	G-SVM	L-SVM	ZERO-TSK-FS	L2-TSK-FS	FCSVM-FS
5	*	*	*	*	*
10		*		*	
15	*	*	*	*	*
20	*	*	*	*	*
25	*	*	*	*	*
30	*	*	*	*	*
35		*	*	*	*
40	*	*	*	*	
45	*	*	*	*	*
50	*	*	*	*	*

* means that there exist significant differences between LR-ZERO-TSK-FS and the state-of-the-art models (*p* < 0.05).

**Table 4 diagnostics-12-02613-t004:** Classification performance in terms of training macro F1-score.

Number of Features	G-SVM	L-SVM	ZERO-TSK-FS	L2-TSK-FS	FCSVM-FS	LR-ZERO-TSK-FS
5	0.5289	0.5947	0.5674	0.6088	0.5098	0.6145
10	0.6287	0.5792	0.6065	0.6474	0.6045	0.6024
15	0.6159	0.6801	0.6543	0.6612	0.6181	0.6887
20	0.6701	0.6810	0.6803	0.6704	0.6251	0.6912
25	0.6342	0.7031	0.7032	0.6305	0.6831	0.7253
30	0.7801	0.7603	0.7454	0.6504	0.6448	0.7923
35	0.7956	0.7251	0.7002	0.6405	0.6523	0.7664
40	0.7406	0.7602	0.6803	0.7603	0.7503	0.7661
45	0.7510	0.7604	0.7711	0.7303	0.7803	0.8047
50	0.7301	0.7402	0.7600	0.7600	0.7589	0.8232

**Table 5 diagnostics-12-02613-t005:** Classification performance in terms of testing macro F1-score.

Number of Features	G-SVM	L-SVM	ZERO-TSK-FS	L2-TSK-FS	FCSVM-FS	LR-ZERO-TSK-FS
5	0.5512	0.5798	0.5925	0.5923	0.5923	0.5812
10	0.5823	0.6612	0.6487	0.6487	0.6487	0.6256
15	0.6741	0.7361	0.7367	0.7118	0.7123	0.7220
20	0.7019	0.7012	0.7010	0.7012	0.7124	0.7489
25	0.7189	0.7490	0.7102	0.7001	0.7302	0.7491
30	0.7478	0.7731	0.7612	0.7742	0.7781	0.8024
35	0.7476	0.7800	0.7803	0.7732	0.7921	0.8215
40	0.7803	0.7586	0.8137	0.7803	0.7803	0.8510
45	0.7897	0.7898	0.7911	0.8021	0.7923	0.8454
50	0.8005	0.8005	0.8008	0.8009	0.8001	0.8702

**Table 6 diagnostics-12-02613-t006:** CPU seconds all algorithms consumed.

Number of Features	G-SVM	L-SVM	ZERO-TSK-FS	L2-TSK-FS	FCSVM-FS
5	1.76	1.43	2.11	1.98	2.03
10	2.32	2.67	3.33	2.99	2.92
15	2.91	3.04	3.50	3.14	3.24
20	3.64	2.98	4.11	4.04	3.96
25	7.21	6.23	5.55	5.98	6.01
30	7.87	6.65	6.04	7.13	7.27
35	8.03	7.75	7.56	8.09	7.85
40	10.43	9.84	7.99	9.43	8.43
45	13.67	13.48	16.09	15.43	16.22
50	16.47	15.99	16.73	17.48	17.78

**Table 7 diagnostics-12-02613-t007:** Optimal features for model training.

No.	Feature Name	No.	Feature Name
1	‘wavelet-HH_glrlm_GrayLevelVariance’	16	‘logarithm_gldm_GrayLevelVariance’
2	‘squareroot_glrlm_GrayLevelNonUniformityNormalized’	17	‘wavelet-HL_glrlm_GrayLevelNonUniformityNormalized’
3	‘exponential_firstorder_RobustMeanAbsoluteDeviation’	18	‘wavelet-HL_ngtdm_Busyness’
4	‘gradient_glcm_ClusterProminence’	19	‘squareroot_glszm_LargeAreaEmphasis’
5	‘squareroot_glcm_MCC’	20	‘squareroot_glrlm_LongRunLowGrayLevelEmphasis’
6	‘squareroot_glcm_ClusterProminence’	21	‘wavelet-LH_glrlm_RunEntropy’
7	‘gradient_glszm_GrayLevelVariance’	22	‘wavelet-LH_glrlm_GrayLevelNonUniformityNormalized’
8	‘wavelet-LH_gldm_LowGrayLevelEmphasis’	23	‘wavelet-HH_gldm_DependenceNonUniformityNormalized’
9	‘exponential_firstorder_Range’	24	‘exponential_glszm_LargeAreaHighGrayLevelEmphasis’
10	‘wavelet-LH_glszm_SmallAreaEmphasis’	25	‘square_glcm_DifferenceEntropy’
11	‘square_gldm_DependenceNonUniformityNormalized’	26	‘original_glcm_SumSquares’
12	‘gradient_glcm_JointAverage’	27	‘squareroot_glcm_Autocorrelation’
13	‘logarithm_glrlm_LongRunHighGrayLevelEmphasis’	28	‘logarithm_glrlm_LongRunEmphasis’
14	‘wavelet-LH_glrlm_LongRunLowGrayLevelEmphasis’	29	‘wavelet-HH_firstorder_90Percentile’
15	‘wavelet-LH_glszm_HighGrayLevelZoneEmphasis’	30	‘wavelet-LH_glrlm_RunEntropy’

**Table 8 diagnostics-12-02613-t008:** Five trained fuzzy rules.

LR-ZERO-TSK-FS		
$k\text{-}\mathrm{th}\;\mathrm{fuzzy}\;\mathrm{rule}:\;\mathrm{If}\;x_{i1}$ $\;\mathrm{is}\;\vartheta_{1}^{k}$ ∧ $x_{i2}$ $\;\mathrm{is}\;\vartheta_{2}^{k}$ ∧ …∧ $x_{id}$ $\;\mathrm{is}\;\vartheta_{d}^{k}$ $,\;\mathrm{then}\;f^{k}(x_{i})=\beta^{k},k=1,2,\ldots ,K.$		
No.	Antecedent	Consequent
1	[0.3888, 0.4187, 0.0043, 0.3884, 0.2543, …] [0.0080, 0.0051, 0.0011, 0.0079, 0.0034, …]	[0.0340, 0.0087, 0.0024]
2	[0.4791, 0.5709, 0.0008, 0.4801, 0.2732, …] [0.0214, 0.0105, 0.0005, 0.0213, 0.0068, …]	[0.0096, 0.0148, 0.0119]
3	[0.2726, 0.4477, 0.0018, 0.2746, 0.2200, …] [0.0199, 0.0088, 0.0007, 0.0198, 0.0050, …]	[0.0100, 0.0108, 0.0077]
4	[0.3443, 0.2711, 0.0072, 0.3471, 0.1323, …] [0.0156, 0.0104, 0.0015, 0.0153, 0.0042, …]	[0.0048, 0.0189, 0.0202]
5	[0.6174, 0.3593, 0.0018, 0.6172, 0.1463, …] [0.0283, 0.0094, 0.0007, 0.0279, 0.0048, …]	[0.0026, 0.0109, 0.0165]

**Table 9 diagnostics-12-02613-t009:** Five trained fuzzy rules in terms of “If…then…”.

Rule No.	If Part	Then Part
1	*No.1 is “low”, No.2 is “low”, No.3 is “a little low”, No.4 is “a little low”, No.5 is “a litter low”, …*	$f^{1}(x)=0.0340$*for class 1,*$f^{1}(x)=0.0087$*for class 2 and*$f^{1}(x)=0.0024$*for class 3*.
2	*No.1 is “low”, No.2 is “low”, No.3 is “a little low”, No.4 is “a little low”, No.5 is “a little low”, …*	$f^{2}(x)=0.0096$*for class 1,*$f^{2}(x)=0.0148$*for class 2 and*$f^{2}(x)=0.0119$*for class 3*.
3	*No.1 is “low”, No.2 is “low”, No.3 is “a little low”, No.4 is “a little low”, No.5 is “a little low”, …*	$f^{3}(x)=0.0100$*for class 1,*$f^{3}(x)=0.0108$*for class 2 and*$f^{3}(x)=0.0077$*for class 3*.
4	*No.1 is “low”, No.2 is “low”, No.3 is “a little low”, No.4 is “a little low”, No.5 is “a litter low”, …*	$f^{4}(x)=0.0048$*for class 1,*$f^{4}(x)=0.0189$*for class 2 and*$f^{4}(x)=0.0202$*for class 3*.
5	*No.1 is “low”, No.2 is “low”, No.3 is “a little low”, No.4 is “a little low”, No.5 is “a little low”, …*	$f^{5}(x)=0.0026$*for class 1,*$f^{5}(x)=0.0109$*for class 2 and*$f^{5}(x)=0.0165$*for class 3*.

## Data Availability

The data are available on: https://github.com/agchung/Figure1-COVID-chestxray-dataset (accessed on 8 August 2021); https://github.com/agchung/Actualmed-COVID-chestxray-dataset (accessed on 8 August 2021); https://www.kaggle.com/tawsifurrahman/covid19-radiography-database (accessed on 8 August 2021); https://www.kaggle.com/c/rsna-pneumonia-detection-challenge/data (accessed on 8 August 2021); Code availability: The code can be accessed by sending email to the corresponding author.

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
