# Peer review of "Radiomics-Based Detection of COVID-19 from Chest X-ray Using Interpretable Soft Label-Driven TSK Fuzzy Classifier"

_diagnostics, 2022, doi:10.3390/diagnostics12112613_

Round 1

Reviewer 1 Report

Through this research paper, the authors have come up with a strong narrative and a very robust algorithm for detecting COVID-19 related CXR abnormalities thus helping early detection and isolation/treatment of suspected cases. This is very critical tool in medical diagnostics keeping in mind the severity of the current pandemic situation. It’s true that although deep learning models have promising performance in detecting these abnormalities, they are regarded as black box in terms of model interpretability due to use of incomprehensible deep features. In such a situation having an equivalent methodical approach which is highly interpretable and having transparency in training procedure is truly an asset in the hands of medical practitioners and researchers.

All in all, I found their approach quite impactful, using critical and rigorous mathematical models to support their claims and observations. I would like to point to few points, which I feel if taken care can make their research and presentation more effective.

11)      As the north star metric, the authors measured the model accuracy and plotted it for all the comparisons. This works fine if the data is balanced as in their case of 200 CXR for each of Covid 19, normal and pneumonia. It would be great if they can use other metrics like precision or recall or F1-score to mark the performance of the models. Tracking performance using these metrics becomes critical and useful when the data is imbalanced in real life situations.

22)      The authors did mention regarding time complexity in the algorithm section 3.3 which is great. At the same time, it would be useful for the readers if the authors mention regarding the system requirements for the machines or processors that they have used for running the algorithm, like the number of cores needed, the specification of the machines used, did they use any Spark clusters or not etc.

33)      Also, there is no mention regarding the processing time. It would be good to have the processing time or running time for the algorithm.

44)      I did not see any link to the package or library or software of the model the authors developed, which other researchers can use to test their CXR images. Do the authors plan to have a package/library in future which can be used by researchers or medical professionals for decision making?

55)      The authors did mention using the radiomics package to extract different types of radiomics features in line 171. I am curious are they referring to the radiomics package in R or pyRadiomics package in Python or any other package? Also, it would be good to know which language have they implemented their algorithm? Is it in Python or R or C/C++ or MatLab or any other language? I don’t see any mention in the manuscript.

66)      The authors mention the time complexity for their proposed model in line 160 which is great!! It is square in “d” which is the feature dimension. My only concern is,  generally in radiomics feature extraction algorithms, 100’s of features are extracted. Like the pyradiomics package extracts ~1500 features. Such high dimensional feature space would lead to very high time complexity due to d-squared in the expression. Are multiple cores or Spark clusters used for running the analysis/models? Although in figures 3 and 5, the authors have shown up to 50 features (I guess they are the top features coming from figure 2 workflow). So, for using up to 50 features, it would be good if the authors can mention how much time the algorithm is taking to run the model with these 50 top features. (this is related with point 3 above)

77)      Some quick question with words used:

a.       The authors have used “tabel” instead of “table” in line 83 and also in table 7.  Is it a typo because I am not aware and also did not find suitable word or meaning for “tabel”.

b.       The authors are using the word “little low”, “little high” in line 263 and also in Figure 7 which is fine and makes sense. But in Table 6, they are using the word “litter low”. I am not too sure I understand what they want to mean by “litter low”?

Rest all looks good. Please let me know if any further clarifications or explanations are required regarding any of the points mentioned above. Congratulations and best wishes to the authors for their innovative approach and effective models.

Author Response

Dear Reviewers,

First of all, we would like to thank the reviewers for all their comments and suggestions, which are very important and valuable in lifting up the quality of the manuscript. We have thoroughly revised the manuscript according to the comments received. The main changes are highlighted in the revised manuscript and the details in response to the comments are attached. Thanks a lot in advance for reviewing the manuscript again.

Best Regards,

Prof.Zhang

Reviewer 2 Report

Paper is interesting and reasonably written. Intro and Refs are adequate, Methods are exhaustive and Results are fair. 

Besides the study limitations already mentioned by the authors themselves, there are a few issues I'd like to point out for authors' comments:

1. The introduced novelty is somehow limited, both in the methods and in the results. The authors should point out better why their contribution is substantially new and not just incremental w.r.t. the state of the art.

2. Validation is somehow lacking: applying the new methodology on another dataset, or using part of the the original one as an independent validation set would help better supporting the authors' claims.

3. The authors stress the interpretability aspects of the new model, but such interpretability is not very well evidenced in the manuscript. I would recommend expanding the discussion part about this point.

4. Also reproducibility is not well discussed - is the CV enough to warrant repeatability of the study? Or a deeper resampling need to be enforced?

5. The considered dataset mix data from different sources, and the homogeneity of the resulting set is far from being granted. Authors are confident that no bias is introduced by such choice?

6. Using Matthews Correlation Coefficient as the elective metric for classifier performance (even multiclass) would be a much better choice.

7.  The proposed model is relying on the extracted radiomics features: which geometric properties are represented by these features? Why not trying a mixed model with both radiomics and features derived by the deep model? 

Finally, a rather stylistic annotation: Method section is very technical, including all mathematical details of the introduced algorithm (plus some well known preliminary material). If on one side this is a well appreciated practice to guarantee transparency, it may prevent part of the potential audience to read the manuscript. I would suggest moving most of the math to a dedicated Appendix, keeping only the key passages in the main text and adding an extended verbal description of the underlying mechanism driving the computations. 

Author Response

(The authors gave the same response as above.)
